# Prediction of Potential Commercially Available Inhibitors against SARS-CoV-2 by Multi-Task Deep Learning Model

**DOI:** 10.3390/biom12081156

**Published:** 2022-08-21

**Authors:** Fan Hu, Jiaxin Jiang, Peng Yin

**Affiliations:** Guangdong-Hong Kong-Macao Joint Laboratory of Human-Machine Intelligence-Synergy Systems, Shenzhen Institutes of Advanced Technology, Chinese Academy of Sciences, Shenzhen 518055, China

**Keywords:** deep learning, drug discovery, multi-task learning, protein–ligand interaction, SARS-CoV-2

## Abstract

The outbreak of COVID-19 caused millions of deaths worldwide, and the number of total infections is still rising. It is necessary to identify some potentially effective drugs that can be used to prevent the development of severe symptoms, or even death for those infected. Fortunately, many efforts have been made and several effective drugs have been identified. The rapidly increasing amount of data is of great help for training an effective and specific deep learning model. In this study, we propose a multi-task deep learning model for the purpose of screening commercially available and effective inhibitors against SARS-CoV-2. First, we pretrained a model on several heterogenous protein–ligand interaction datasets. The model achieved competitive results on some benchmark datasets. Next, a coronavirus-specific dataset was collected and used to fine-tune the model. Then, the fine-tuned model was used to select commercially available drugs against SARS-CoV-2 protein targets. Overall, twenty compounds were listed as potential inhibitors. We further explored the model interpretability and exhibited the predicted important binding sites. Based on this prediction, molecular docking was also performed to visualize the binding modes of the selected inhibitors.

## 1. Introduction

As of June 2022, SARS-CoV-2 has sickened more than five hundred million and killed over six million people across the globe. SARS-CoV-2 is the seventh member of the family of coronaviruses that infect humans [1,2]. Similar to MERS-CoV and SARS-CoV, SARS-CoV-2 causes severe respiratory diseases and is capable of spreading from person to person. Unfortunately, the rapid mutation of this virus makes it difficult to develop effective vaccines, especially for Omicron. The COVID-19 pandemic will last for years. Therefore, the continuous development of effective anti-SARS-CoV-2 drugs is necessary to prevent the worsening of symptoms, or even death. As a positive-sense, single-stranded RNA beta-coronavirus, SARS-CoV-2 encodes structural, non-structural and accessory proteins. Among these, RNA-dependent RNA polymerase (RdRp), 3-chymotrypsin-like protease (3CLpro), papain-like protease (PLpro), helicase and the spike glycoprotein are supposed to be the main targets. Several compounds that targeted these viral proteins and inhibited coronavirus in vitro have been reported and moved into clinical trials [3]. For example, remdesivir is an approved HIV reverse-transcriptase inhibitor, which has broad-spectrum activities against RNA viruses such as MERS-CoV and SARS-CoV. However, it showed less effective activity in an Ebola clinical trial [4,5,6]. Previous reports showed that remdesivir inhibited SARS-CoV-2 in vitro with an EC50 of 0.77 μM [7], and was used to treat a SARS-CoV-2-infected patient in the United States [8]. However, more potential inhibitors against SARS-CoV-2 are still needed.

Computational methods, such as molecular docking, select compounds that can bind to a target protein and thus improve the success rates of drug discovery. Recently, methods based on deep learning have gained impressive performance in terms of protein–ligand binding prediction [9,10,11]. One main advantage of this algorithm is that it can extract hidden features automatically from raw data and thus significantly improve the prediction accuracy. However, deep learning models might suffer from the generalizability issue due to lack of data. Recently, pretraining models on a large-scale dataset before applying to a small dataset has emerged as a powerful paradigm for solving this issue [12].

In this study, we propose a multi-task deep learning model for selecting potential SARS-CoV-2 inhibitors. First, the model was pretrained on several heterogenous protein–ligand interactions datasets (Figure 1). The model achieved competitive results on several protein–ligand benchmark datasets. Next, a coronavirus-specific dataset was collected and used to fine-tune the model. The fine-tuned model was then used to select commercially available drugs against the SARS-CoV-2 targets. Overall, twenty drugs were listed as potential inhibitors. Furthermore, we explored the model interpretability and found that the predicted important binding sites were close to ground truth.

## 2. Results and Discussion

### 2.1. Model Performance on Benchmarks

We first trained a multi-task deep learning model on heterogenous protein–ligand interaction datasets. The model achieved competitive results on several benchmark sets. For example, the PDBbind v.2016 dataset is a widely used benchmark for evaluating protein–ligand interaction prediction methods [15]. It provides more than ten thousand protein–ligand structural complexes with binding affinity (e.g., K_d_, K_i_). We split the PDBbind set the same way as Pafnucy [16], a classic deep learning-based method for predicting protein–ligand affinity. Moreover, two independent test sets CASF-2013 [17] and Astex Diverse [18] were used to test the generalizability of the model.

Two metrics, RMSE (root mean square error) and Pearson′s correlation coefficient R, were used to evaluate the regression task. As shown in Figure 2, the model achieved RMSE = 1.538, R = 0.71 on the test set and performed well on both independent tests. These results are better than most classic structure-based methods. It is worth noting that our model is only sequence-based.

For the classification task, the model was evaluated on DUD-E, *Human*, *C. elegans* and KIBA datasets using 3-fold cross validation. Two traditional machine learning algorithms, SVM and random forest, were also used to compare these datasets. As shown in Figure 3, the proposed model showed excellent performance on most evaluation metrics, including AUC, accuracy, precision, recall, F1-score and specificity. Specifically, the single-task method, which indicates training on one specific label, achieved better performance than the multi-task method on more specific metrics such accuracy and precision, while the multi-task method showed better performance on recall, indicating that it covered more protein–ligand interaction data space. It should be noted that each single-task model corresponded to only one dataset (i.e., it performed well on one dataset but performed poorly on the other datasets). Therefore, the single-task model is likely to perform poorly on coronavirus-specific datasets due to the generalizability issue, even though it performed well on the benchmark dataset. In contrast, the multi-task model was trained once and achieved excellent results on all these datasets. These results suggest that the multi-task model has a better applicability by leveraging heterogenous datasets.

### 2.2. Screening of SARS-CoV-2 Inhibitors

Recently, pretraining a model on a large-scale dataset before applying it to a small dataset has emerged as a powerful paradigm for improving model generalizability. Inspired by this idea, we fine-tuned the pretrained model on a collected SARS-CoV-2 dataset and then selected potential SARS-CoV-2 inhibitors from a bioactive commercial library containing 10 thousand compounds. After excluding drugs with possible side effects, a total of 10 drugs with high binding affinity were identified, as listed in Table 1. Among these, abacavir (sulfate), a powerful nucleoside analog reverse-transcriptase inhibitor used to treat HIV (human immunodeficiency virus) was predicted to have high binding affinity with multiple proteins of SARS-CoV-2 including RdRp and helicase. Darunavir, a protease inhibitor used to treat HIV, was used in a clinical trial against COVID-19 (ChiCTR2000029541). It should be noted that both darunavir and darunavir (ethanolate) were not present in the training and fine-tuning sets. That is, they were “unseen” by the model. Our model also predicted that darunavir could target 3CLpro and PLpro with affinity K_d_ = 57.30 and 46.16 nM, respectively, while darunavir (ethanolate) binds to 3CLpro and PLpro with affinity K_d_ = 44.51 and 35.86 nM, respectively. These results partially prove the accuracy and generalizability of our model. In our predictions, almitrine mesylate, which is a respiratory stimulant that enhances respiration, was used in the treatment of chronic obstructive pulmonary disease. Roflumilast has anti-inflammatory effects and is used as an orally administered drug for the treatment of inflammatory conditions of the lungs such as chronic obstructive pulmonary disease. These two predicted drugs are associated with respiratory symptoms that are the main clinical symptoms of COVID-19. Interestingly, kesuting syrup and keqing capsules were used in a trial for the treatment of mild and moderate COVID-19 (ChiCTR2000029991). It is uncertain whether these drugs only help to alleviate clinical symptoms or have a direct effect on the virus. Daclatasvir is used against Hepatitis C Virus (HCV), which stops HCV viral RNA replication and protein translation by directly inhibiting HCV protein NS5A. In this study, the predicted binding affinity between daclatasvir and RdRp was 15.03 nM. Fiboflapon sodium, a high-affinity 5-lipoxygenase-activating protein inhibitor used for the treatment of asthma, was predicted to have potential affinity to PLpro with K_d_ = 197.63 nM.

Furthermore, the model was applied to screen lead compounds for 3CLpro inhibition from a diverse screening library containing two million compounds. This screening library, namely, the Enamine HTS Collection, encompasses versatile chemotypes developed within a couple of decades of chemical research at Enamine and its partner academic organizations. Compounds within this library frequently have unusual structures and unique properties. 3CLpro was taken as the screening target because of its wide application and large amount of corresponding data. Thus, the fine-tuned deep learning model had relatively higher accuracy for selecting 3CLpro inhibitors. For the screening, the candidate compounds were first screened by our model. Those candidates were ranked according to the predicted scores. Then, candidates with high predicted scores were further screened based on their possible side effects. For example, Panobinostat, which is a potent and orally active non-selective HDAC inhibitor and has antineoplastic activity, was predicted to inhibit RdRp at 111.19 nM. However, Panobinostat has been reported to possibly cause side effects such as severe diarrhea, headache and even severe infection. Thus, these candidates with severe side effects were excluded. Finally, a total of 10 potential lead compounds are listed in Table 2.

### 2.3. Model Interpretation

To explore how the model discerns protein–ligand interaction, we conducted a method to identify the key amino acids that are critical for binding. The listed potential inhibitors were regarded as positive samples with high prediction scores. Then, we masked sub-sequences of samples to obtain a “masked” prediction score, and then calculated the importance of these masked sub-sequences. As shown in Figure 4, the critical parts for binding in protein sequences were visualized using a heatmap. In the heatmap, the brighter the region, the more important it is. For 3CLpro, the important amino acids for binding are mostly located at two main parts. It should be noted that different drugs result in different weights in these two regions. For example, roflumilast has higher weights in the first region, indicating the binding sites for roflumilast are close to the middle pocket of 3CLpro. As for abacavir (sulfate), both regions more or less affect the binding, especially the second region at the 180–200th amino acids. For PLpro, the predicted binding sites are located at the 100–120th amino acids.

Moreover, we visualized the predicted binding sites in 3D structures to show possible pockets. The 3CLpro of SARS-CoV-2 (PDB: 6lu7) was used. As mentioned above, two regions of 3CLpro contributed mostly to the binding. As shown in Figure 4A, the region in the upper part (100–200th amino acids) was predicted as the main pocket due to the high weights in most predictions. SARS-CoV-2 PLpro has a catalytic triad composed of Cys114-His275-Asp289 and a conventional zinc-binding domain of four cysteine residues: Cys192, Cys194, Cys227, and Cys229 [19]. The central “thumb” domain contributes the catalytic Cys112 to the active site. Similarly, our model predicted that the 100–120th amino acids contributed mainly to the final binding of small molecules.

### 2.4. Molecular Docking

Based on our prediction and known protein–ligand structural complexes, we performed the molecular docking of predicted inhibitors to 3CLpro. Four inhibitors, Z56899184, Z57728899, Z57013003 and Z1245218850, were docked to 3CLpro using ArgusLab [20]. For each compound, the best pose with the lowest binding affinity from the ArgusLab docking results was selected. As shown in Figure 5, these inhibitors all bind to amino acids that are located in the active site of 3CLpro (the predicted upper binding region in Figure 4A). Previous studies have shown that the catalytic residue Cys145 is critical for forming covalent bonds between 3CLpro and inhibitors [21,22]. As shown, compound Z56899184 also forms a covalent bond with Cys145 of 3CLpro in the docking result (with binding affinity: −8.67 kcal/mol). These results indicate that Cys145 plays a critical role in the binding process. It should be noted that the binding affinity between the protein and ligand cannot be simply considered based on docking scores (i.e., often expressed in kcal/mol).

## 3. Conclusions

In this work, a multi-task deep learning model was proposed to predict potential commercial drugs against SARS-CoV-2. There are two main points that should be restated. First, the model was pretrained on heterogenous protein–ligand interaction datasets before being applied to the SARS-CoV-2 inhibitor screening. This idea was inspired by the recent success of pretrained models on various fields such as natural language processing. The core logic of this method was transfer learning, which could extract features from large-scale datasets and then transfer the learned knowledge to a small set. This method has a broad application in drug discovery which is often hindered by the small data issue. Although the pretrained model had wide distribution in the protein–ligand interaction data space, the distance between existing knowledge and future application is ambiguous. Therefore, which part of the pretraining set would cause issues such as negative effect was not understood well. Second, the exploration of the “black box” within deep learning model is very important, especially in the drug discovery area. The black box model may increase the risk of subsequentially false leads, thus causing severe consequences. Our model could provide some insights into this. Overall, we are on a cheerful avenue for the application of pretrained deep learning models in drug discovery.

## 4. Methods

### 4.1. Data

A few benchmark protein–ligand interaction datasets were used for model pretraining, including DUD-E [23], *H**uman*, *C. elegans* [24], KIBA [25] and PDBbind [15]. For classification, DUD-E, Human, C. elegans and kinase inhibitor bioactivity (KIBA) were used. A total of 91,220 samples from the DUD-E dataset were used in this study, in which the negative-to-positive ratio was set to 3:1 to avoid an imbalance of data. Similarly, the negative-to-positive ratios of Human and C. elegans sets were also set to 3:1. For the KIBA, protein–ligand interactions with values bigger than 12.1 were regarded as positive samples. For regression, the PDBbind v.2016 and kinase dataset Davis were used. Specifically, 13,196 protein–ligand complexes from the PDBbind set were split the same way as Pafnucy. Similarly, the kinase dataset Davis consisting of a total of 9125 interactions with the corresponding binding affinity (Kd) was used here.

For drug screening, several important SARS-CoV-2 proteins were used as targets, such as RNA-dependent RNA polymerase (RdRp), 3-chymotrypsin-like protease, papain-like protease and helicase. The sequences of these proteins were extracted from NCBI (NC_045512.2). The SARS-CoV/SARS-CoV-2 specific dataset was collected from various papers [21,26,27,28,29] and public datasets such as GHDDI and PubChem with filtration. After removing duplicates, a collection of 10 thousand approved and bioactive compounds was used for screening. Similarly, a diverse Enamine HTS collection containing 2 million compounds was also used.

### 4.2. Model

Basically, the proposed model consists of three parts: protein feature extraction by word2vec and transformer, drug feature extraction by node2vec and interaction prediction by multi-task modules. The last dense layer was activated as output. The loss functions were defined as binary cross entropy (BCE) and mean squared error (MSE) for classification and regression, respectively. Figure 1 displays the basic architecture of the model, and more details are described below.

**Preprocessing.** Given the drug SMILES and protein sequence, we first constructed a graph and words, respectively, and then obtained the dense vectors by the embedding method. Let $L_{idx}=x_{1},\;x_{2},\ldots ,\;x_{L-1},\;x_{L}$ be a word sequence of each protein, where *idx* is the sequence index, *L* is the word sequence length and $x_{i}\in R^{d_{1}}$ is the d1-dimensional embedding of the *i*th word that we split by overlapping 3-gram on the original sequence. For drugs, let G=(V, E) be a given drug SMILES and $f:V\to R^{d_{2}}$ be the mapping function from $V_{idx}=\{v_{1},\;v_{2},\;\ldots ,\;v_{l-1},\;v_{l}\}$ to feature representation. Where idx is the sequence index, l is the drug length and $v_{i}$ is the *d*2-dimensional embedding of the ith node on the original sequence [30].

**Pretraining.** With the building of a large-scale word-segmented corpus of protein sequence, we then used each current word as an input to a log-linear classifier with a continuous projection layer, and predicted words within a certain range before and after the current word, proposed as a skip-gram model [13], to obtain the final $d_{1}$-dimensional embedding words. Meanwhile, to identify the specific drug network nodes, we considered node2vec [14] as the representation learning method to design a flexible neighborhood sampling strategy to explore a random walk, such as $\left\{s,\;v_{3},\;v_{5},\;v_{9},\;v_{8},\;v_{1},\;v_{6},\;v_{4}\right\}\;$ of start node $s=v_{2}$ and walking length *l* = 8; thus, we can consider the skip-gram architecture mentioned above as a feature learning method.

**Multi-task module.** This module consists of two parts: shared layers and task-specific layers. The shared layers are designed to learn a joint representation for all tasks. Task-specific layers are used to learn the weights of specific blocks based on the joint representation. In this study, two related tasks are defined: classification and regression. The model was fine-tuned by a virus-specific dataset to acquire robust results for coronavirus. 

The model weights were initialized using a Glorot uniform initializer, which keeps the variance of output and input by correcting the variance of uniform. Then, the Adam optimize was used to train the model. The initial learning rate and batch size was set to 10-4 and 256. During the training process, early stopping was used to monitor the training and validation loss in order to select the final model without overfitting.

### 4.3. Biological Interpretation

A non-parametric method, “occlusion”, used in our previous study [11], was applied to explore which parts of the input protein sequences were critical to the task. The intuitive explanation of this method is: (i) mask a subsequence (e.g., 10–20th amino acids) of a full-length protein sequence; (ii) predict a result of this modified sequence; (iii) calculate the difference between this result and the ground truth; (iv) repeat (i) to (iii) and find the most important amino acids. Specifically, *s_i_* from test samples (*i* = 0, 1, 2,..., n−1, here, n is sample size of test set) was expressed as tuple (protein input_*i*, compound input_*i*). While maintaining compound input_i as unchanged, we systematically masked the protein input_i in *s_i_* to track the changes of the output. Then, the importance of each sub-sequence in the sequence to the prediction could be calculated.

## Figures and Tables

**Figure 1 biomolecules-12-01156-f001:**
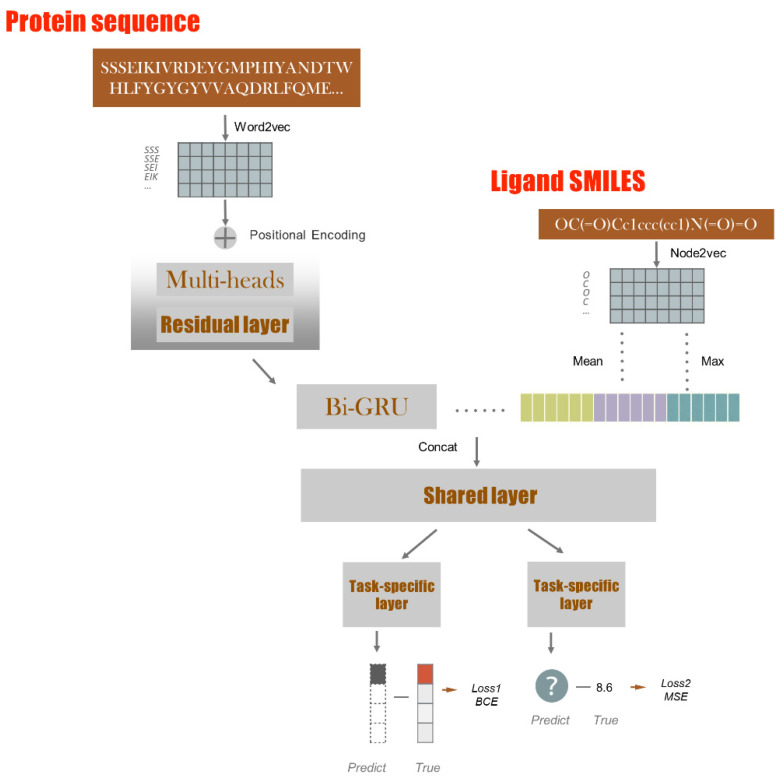
Schematic of the proposed model. The model involves two parts: protein/ligand feature extraction and their interaction prediction. First, the protein sequence is processed in turn by word2vec [13], multi-heads residual layer and Bi-GRU (bidirectional gated recurrent unit) modules. Ligand smiles is processed by node2vec [14]. Then, their representations are fed into a shared layer and task-specific layer and connected to the labels.

**Figure 2 biomolecules-12-01156-f002:**
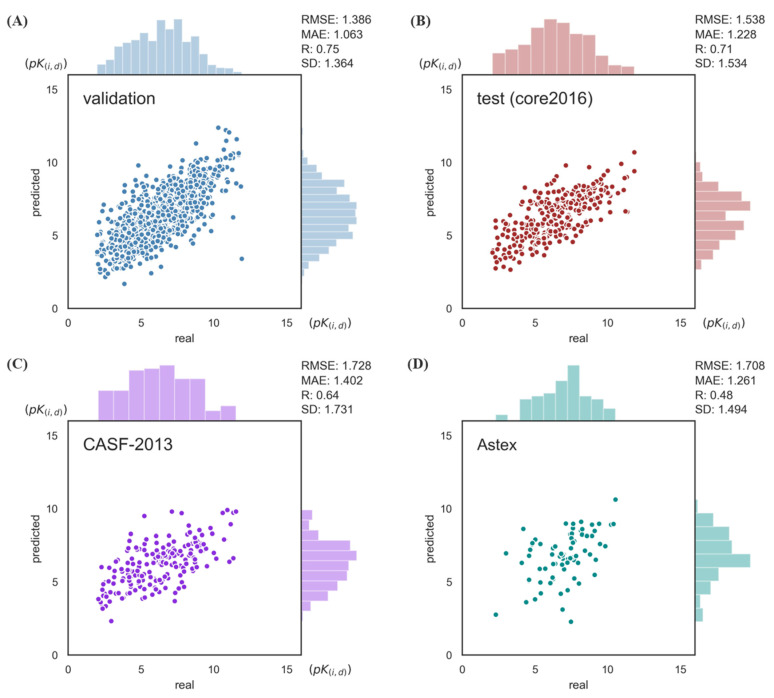
Model performance on the PDBbind v.2016 set and two independent sets. (**A**) PDBbind validation set. (**B**) PDBbind v.2016 core set. (**C**) CASF-2013. (**D**) Astex Diverse set. RMSE: root mean square error. MAE: mean absolute error. R: Pearson′s correlation coefficient R. SD: stand deviation.

**Figure 3 biomolecules-12-01156-f003:**
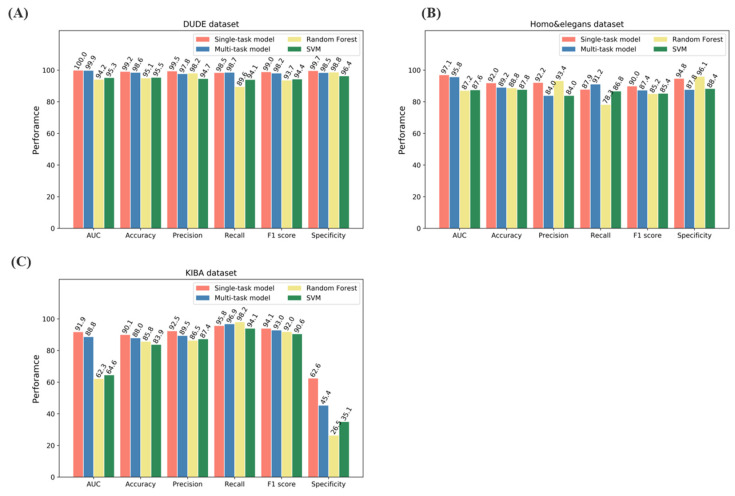
Model performance on (**A**) DUD-E. (**B**) *Human* and *C. elegans.* (**C**)KIBA datasets.

**Figure 4 biomolecules-12-01156-f004:**
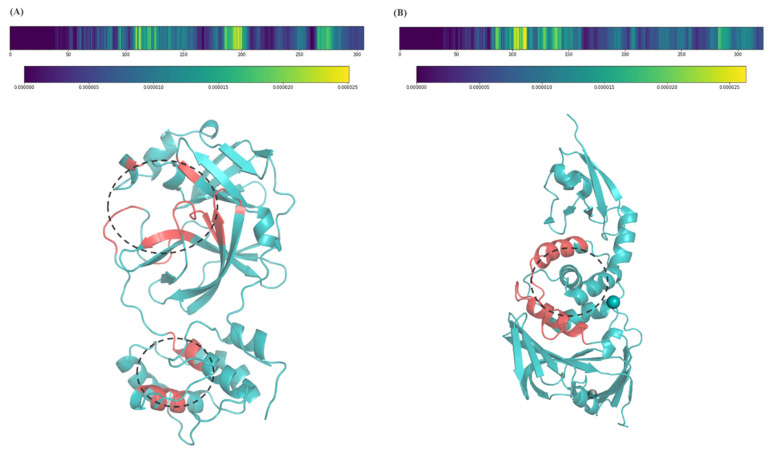
The predicted binding sites. (**A**) 3CLpro (PDB ID: 6lu7); (**B**) PLpro (PDB ID: 6wx4). The abscissa axis is the length of protein sequences. Red indicates the predicted important sites and black dotted circles indicate possible pockets.

**Figure 5 biomolecules-12-01156-f005:**
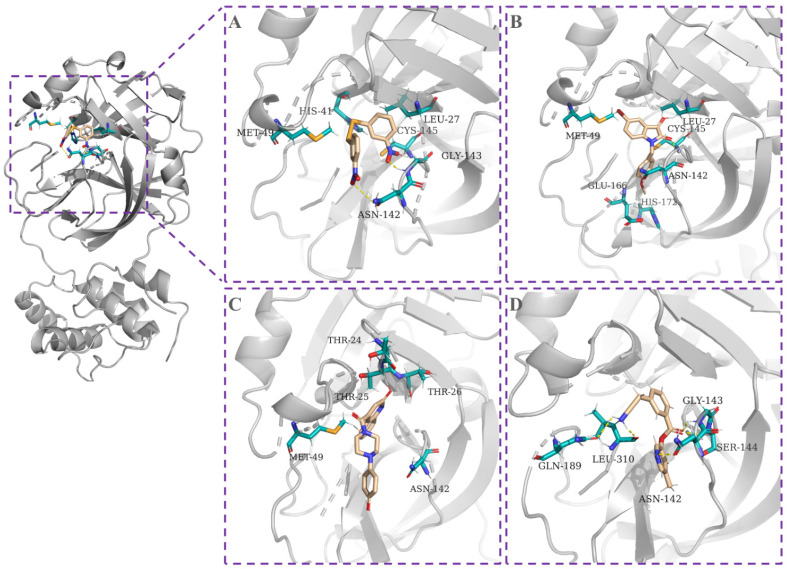
Molecular docking of predicted inhibitors to SARS-CoV-2 3CLpro. (**A**) Z56899184, −8.67 kcal/mol; (**B**) Z57728899, −7.86 kcal/mol; (**C**) Z57013003, −7.45 kcal/mol; (**D**) Z1245218850, −8.41 kcal/mol. 3CLpro is represented as translucent gray cartoon. The involved amino acids are represented in cyan.

**Table 1 biomolecules-12-01156-t001:** Potential inhibitors for SARS-CoV-2.

Drug	CAS	Target	Predicted Affinity (nM)
Abacavir (sulfate)	188062-50-2	RdRp helicase	3.03 3.06
Darunavir	206361-99-1	3CLpro PLpro	57.30 46.16
Darunavir (ethanolate)	635728-49-3	3CLpro PLpro	44.51 35.86
Itraconazole	84625-61-6	PLpro RdRp	127.98 16.90
Almitrine mesylate	29608-49-9	3CLpro	29.31
Daclatasvir	1009119-64-5	RdRp	15.03
Daclatasvir (dihydrochloride)	1009119-65-6	RdRp	19.87
Metoprolol tartrate	56392-17-7	PLpro	153.23
Fiboflapon sodium	1196070-26-4	PLpro	197.63
Roflumilast	162401-32-3	3CLpro	248.89

**Table 2 biomolecules-12-01156-t002:** The predicted lead compounds for SARS-CoV-2 3CLpro.

ID	SMILES	Probability
Z56899184	O=[N+]([O-])c1cccc(SSc2cccc([N+](=O)[O-])c2)c1	0.969
Z229622170	N#Cc1cccc(CN2C(=O)C(=O)c3cccc(Br)c32)c1	0.947
Z57728899	COc1ccc(CN2C(=O)C(=O)c3cc(Br)ccc32)cc1	0.918
Z90667629	[Na+].[O-][n+]1ccccc1[S-]	0.863
Z1238998507	N#Cc1cc([N+](=O)[O-])ccc1Oc1cncc(Cl)c1	0.782
Z56833036	O=C1c2ccccc2C(=O)c2c1cccc2S(=O)(=O)N1CCOCC1	0.724
Z57013003	COc1ccc(N2CCN(C(=O)c3cc(=O)[nH]c4ccccc34)CC2)cc1	0.711
Z1245218850	N#CCc1cccc(C(=O)Oc2cncc(Cl)c2)c1	0.697
Z56785091	c1nc(SSc2nc[nH]n2)n[nH]1	0.682
Z1776036493	Cc1cc(Br)cc2c1N(CCBr)C(=O)C2=O	0.676

## Data Availability

The source code and data are available at https://github.com/AI-bio/multi-task-for-cov, accessed on 17 August 2022.

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
