# Peer review of "Prediction of Potential Commercially Available Inhibitors against SARS-CoV-2 by Multi-Task Deep Learning Model"

_biomolecules, 2022, doi:10.3390/biom12081156_

Round 1

Reviewer 1 Report

Manuscript Title: Prediction of potential commercially inhibitors against SARS-CoV-2 by multi-task deep learning model

Reviewer Recommendation: Major Revision

Comments

1. Authors have mentioned that a total of 271,816 protein-ligand interactions were selected after preprocessing; what are the important criteria that have been used in preprocessing to screening the protein-ligand interaction?

2. Authors have mentioned that around 10 thousand approved bioactive molecules were selected from literature and PubChem database; are all the compounds antiviral compounds?

3. Do authors have focused only on specific activity of the compound to screen the compounds from literature and pubchem databases?

4. In line number 172, it was mentioned that Word2vec but in line number 173, it was noticed that node2vec; is these are two different model or typographical errors. Make it clear. 

5. Authors have mentioned that from the 10 thousand compounds 10 compounds were selected by excluding drugs with possible side effects; is this the only one criteria used for screening the compounds?

6. Authors have requested to provide the information on what are the important side effects that are majorly associated with the compounds those are excluding from this screening.

7. The detailed abbreviations of some short forms are not provided in the manuscript and it makes reading the manuscript difficult. 

8. In table 1 authors have mentioned that the binding affinity of the compounds in mM concentration; is the binding affinity of the compounds was experimentally proved: if so provide information on what kind of bio-assay has been used in this study?

9. The following sentence is difficult to understand. Rewrite it.

“The yellower in color the region, the more important it is.”

Reviewer 2 Report

The manuscript proposes a series of  SARS CoV 2 inhibitors retrieved from commercially available sources using a deep learning model.

Abstract: OK

Introduction: Nobody knows when the pandemic will finally end  - please rewrite.

From the PDBbind v 2016, what database is used? ( protein-ligand complex, nucleic acid ligand complex? Please be more specific.

 Please describe and add the two data sets to supplemental files. (Pafnucy, CASF-2013)

PDB bind includes general data regarding protein interaction; how is this applied explicitly on ligand SARS CocV interaction?

Describe the training set and test set. Please add this file to supplemental materials.

Please add a chart of the model validation test set vs. the predicted set.

Regarding the ZN database, please describe the database. For example, how many compounds were used for screening?

Line 115, please be more specific about the methodology.

Molecular docking – 1 How the binding sites were revived

What are the binding site coordinates, and how was the docking validated?

Conclusions: please re write conclusion ( the conclusions should not be written as an abstract)

 Methods:

Data: please be more specific about the data set.

Model: Please state precisely the step how the model was built and not general terms ( which could be applied universally)

Biological interpretation: please be more specific about what exactly was performed.

Reviewer 3 Report

Comments on biomolecules-1849831:

The paper reports a deep-learning screening and docking calculation of inhibitors against SARS-CoV-2. The goal of the work would be interesting to readers, but the presentation and the scientific solidity require significant improvements. There are also many formatting issues due to careless proofreading, which should be corrected before submission of the manuscript. 

The main part of this study is the training and application of a multi-task deep learning model. However, the code is not provided in the supporting information or uploaded to a public repository, which hinders the greatest use to the community.

The paper only reports quality measures indicating the overall performance of the method on several datasets. However, the detailed predictions for specific systems remain unreported. These data should be included somewhere online (available to readers) or provided in the supporting information.

The error of the prediction is rather large (RMSE ~1.5 for pKd or pKi). What is the reason for this behavior? Is there any technique that could improve the results? Could the authors do some efforts on this aspect by modifying the details of their model to better the performance, rather than simply constructing a model without justification?

Formatting: There are many abbreviations directly used without explicit definitions. For instance, MAE and SD show up in Fig. 2 without explanation. This is quite reader-unfriendly and should be corrected before submission. 

Author list. In the online submission system, the author Fan Hu is listed as the corresponding author, while in the submitted manuscript Peng Yin is assigned as the corresponding author. Some clarification on this aspect should be added.

The authors select two pdb structures in their modelling, i.e., 6lu7 and 6wx4. Is there any specific and scientifically solid reason for this selection?

Molecular docking is performed on the screened ligands (Z56899184, Z57728899, Z57013003 and Z1245218850), all of which are found to have favorable binding affinities towards the 3CLpro target. However, I wonder whether this step is really useful, as the design of this investigation and the solidity of the small-sample-size docking calculation are questionable. If the authors enlarge the set of ligands considered in docking, e.g., including more ligands that are predicted to be less potent in previous steps, the docking calculation would also produce favorable docking scores and even more potent than the existing four inhibitors, the reasons for which include the inaccuracy of the scoring function, the lack of exhaustiveness of conformational search in molecular docking, and the errors of the deep learning model. As the screened dataset of inhibitors is not really large, why not using more rigorous free energy techniques to confirm the results?  

The authors mention covalent binding to CYS145 but perform non-covalent docking calculations for the obtained inhibitors, which also seems strange in their screening workflow. There are many tools to do this covalent-binding docking job, e.g., Arguslab and Argusdock. The docking step should really provide useful information and such covalent type calculation is necessary. Make it a solid work!

A further comment on molecular docking is the aim of this calculation. The authors state that docking is performed to visualize the binding modes of selected inhibitors. However, it is widely acknowledged that docking calculations often lead to unreliable predictions of binding modes, which often require post-docking atomistic simulations to refine and even perform a totally new conformational search for reliable descriptions of binding pose. Therefore, the docking calculation reported in this paper seems to be designed to achieve a goal beyond its capabilities, which does not seem reasonable.

I search for the paper online and find a preprint version at https://arxiv.org/abs/2003.00728. In that case, a preprint acknowledgement should be added.

Round 2

Reviewer 1 Report

Revised version can be accepted for publication, 

Best wishes for authors

Author Response

We thank the reviewer for taking the necessary time and effort to review the manuscript.

Reviewer 2 Report

Point1: OK

Point2: please add a link or data set to supplemental files

Point 3: OK

Point4: OK

Point5: Please add at least 1% of screening results to the supplemental files ( preferably in the order of docking scores)

Point 6: Please add the response to the manuscript.

Point 7: Is this the center of the molecule?

Point 8: OK

Point 9: Please add the response to the manuscript text

Point10: OK

Point11: please elaborate more on this response and add it to the manuscript.

The authors responded in some parts to the comments. However, there are still some issues that must be addressed before the manuscript can be published.

Reviewer 3 Report

Acceptable in the current form. 

Author Response

(The authors gave the same response as above.)
